# Redistribution of TNF Receptor 1 and 2 Expression on Immune Cells in Patients with Bronchial Asthma

**DOI:** 10.3390/cells11111736

**Published:** 2022-05-24

**Authors:** Alina Alshevskaya, Julia Zhukova, Fedor Kireev, Julia Lopatnikova, Irina Evsegneeva, Daria Demina, Vera Nepomniashchikch, Victor Gladkikh, Alexander Karaulov, Sergey Sennikov

**Affiliations:** 1Federal State Budgetary Scientific Institution, Research Institute of Fundamental and Clinical Immunology (RIFCI), Novosibirsk 630099, Russia; alkkina@yandex.ru (A.A.); zhukova1982@rambler.ru (J.Z.); f.kireev@mail.ru (F.K.); lopatnikova_j_a@ngs.ru (J.L.); immunology@mail.ru (D.D.); niiki_imm@mail.ru (V.N.); 2Federal State Autonomous Educational Institution of Higher Education I.M. Sechenov, First Moscow State Medical University of the Ministry of Health of the Russian Federation, Moscow 101000, Russia; ivevsegneeva@yandex.ru (I.E.); drkaraulov@mail.ru (A.K.); 3Biostatistics and Clinical Trials Center, Novosibirsk 630099, Russia; gladvs_ru@mail.ru

**Keywords:** TNF-alpha, bronchial asthma, cellular immunology, cytokine receptors expression, immune regulation

## Abstract

Background: The co-expression patterns of type 1 and 2 tumor necrosis factor (TNF)-α membrane receptors (TNFR1/TNFR2) are associated with the presence, stage, and activity of allergic diseases. The aim of this study was to assess the expression levels and dynamics of TNFRs on immune cells and to assess associations between their expression and severity of bronchial asthma (BA). Methods: Patients with severe (n = 8), moderate (n = 10), and mild (n = 4) BA were enrolled. As a comparison group, data from 46 healthy volunteers (HV) were accessed. Co-expression of TNFR1/2 was evaluated as a percentage of cells and the number of receptors of each type per cell. Multivariate logistic regression analysis was used to identify diagnostic biomarkers of BA. Results: More than 90% of the monocytes in patients with mild BA were TNFR1+TNFR2+ but had significantly lower TNFR1 expression density compared with HV (7.82- to 14.08-fold, depending on disease severity). Lower percentages of the TNFR+ B-lymphocytes were observed in combination with significantly lower receptors density in BA compared with HV (2.59- to 11.64-fold for TNFR1 and 1.72- to 3.4-fold for TNFR2, depending on disease severity). The final multivariate model for predicting the presence of BA included the percentage of double-positive CD5+ B-lymphocytes and average number of TNFR1 molecules expressed on cytotoxic naive T-lymphocytes and T-helper cells (R^2^ = 0.87). Conclusions: The co-expression patterns of TNFRs on immune cells in BA differed significantly compared with HV. The expression differences were associated with disease severity. TNFR1 expression changes were key parameters that discriminated patients with BA from those with HV.

## 1. Introduction

Because of the broad range of its biological functions, the cytokine tumor necrosis factor (TNF)-α plays a crucial role in immune regulation both in healthy individuals and in patients with various diseases [1,2]. Earlier studies showed that TNF-α can regulate both proinflammatory and anti-inflammatory processes [3]. TNF-α functions by transducing signals through two types of TNF receptors (TNFR1 and TNFR2). TNFR1 signaling is predominantly involved in cytotoxic, proinflammatory, and apoptotic processes [1], while TNFR2 signaling predominantly mediates the activation and proliferation of mononuclear cells [4,5]. However, complex patterns of regulation and cross-talk involving TNFR co-expression and co-activation have also been demonstrated [6,7].

The role of TNF-α and its receptors in the pathogenesis of bronchial asthma (BA) has been extensively studied both at local [8,9] and systemic levels [10,11,12,13]. BA is accompanied by TNFR1-mediated NF-κB overactivation, leading to increased expression of numerous inflammatory genes in patients with BA and the induction of airway inflammation [14]. By contrast, TNFR2-mediated activation of signal transduction reduces inflammatory infiltration in the airways [15] via reduced differentiation of T helper (Th) 2 and Th17 cells, as well as decreased cytokine expression in serum and bronchoalveolar lavage fluid [16].

These data demonstrate that the rate of disease development in patients with BA may be largely regulated by the balance regarding TNFR signaling in immune cells. The cellular response to cytokines is regulated through several mechanisms, including cytokine levels (in the soluble and membrane-bound forms), receptor expression level on cells, and the ratio between different types of receptors in each cell subset [7,17,18].

Cytokine receptor expression levels differ among immune cell subsets, and expression patterns change significantly in patients with autoimmune, allergic, and infectious diseases [19,20,21]. Based on the established associations between changes in the co-expression of TNFRs and pathologic conditions, we hypothesized that the expression and co-expression patterns of TNFR1 and TNFR2 could serve as diagnostic tools to assess the disease severity and predict the response to therapy in patients with BA. Therefore, we aimed to evaluate the co-expression patterns and the average number of TNFR1 and TNFR2 receptors in patients with BA compared with those of healthy volunteers. Our overall goal was to identify disease-specific indicators of the presence and severity of BA.

## 2. Methods

Whole blood samples from healthy volunteers and patients with BA undergoing treatment at the Department of Allergology, Immunopathology Clinic of the Research Institute of Fundamental and Clinical Immunology (Novosibirsk, Russia) were obtained for assessment of TNFR1/TNFR2 co-expression. A total of 46 healthy volunteers aged 18 to 77 years (median, 36.5 years; interquartile range (IQR), 30 to 54 years) were enrolled in the study: 16 men (34.8%) and 30 women (65.2%). Selection of healthy volunteers was carried out after basic check-up, including an assessment of cardiac and respiratory functions. A total of 22 patients with allergic form of BA aged 22 to 70 years (median, 46 years; IQR, 32 to 49 years) were enrolled in the study: eight men (27.3%) and 16 women (72.7%). The criterion for inclusion of patients in the study was the absence of other immune-mediated conditions and diseases that could affect the redistribution of receptors on the surface of immunocompetent cells. The study was approved by the local ethics committee of RIFCI (protocol no. 24, dated 8 September 2016). All patients and healthy volunteers provided written informed consent for study participation and for publication. Table 1 shows the baseline demographic and clinical characteristics of the healthy volunteers and patients with BA receiving different courses of treatment. Spirometry was carried out in the period from 8.00 to 9.00 a.m. with spirograph MAC2-BM (Belintelmed, Minsk, Belarus). The instrument was calibrated daily using a 3 L calibration syringe. Before the study, the patient’s anthropometric data (gender, age, height, and weight) were filled in to the program; data on smoking were indicated. The quality criteria during the procedure were automatically controlled by the instrument. Vital capacity and forced vital capacity were evaluated in at least three attempts. After the tests were completed, the computer program calculated the average parameters of the respiratory function, including forced expiratory volume, in one second (FEV1).

Venous blood samples (6 mL) were collected from the ulnar vein under fasting conditions into vacuum tubes containing ethylenediaminetetraacetic acid (Greiner Bio-One GmbH, Kremsmünster, Austria) using aseptic technique. Blood samples were collected from patients at admission to the hospital before the correction of basis therapy. Prior to hospitalization, patients received basic therapy for at least 6 months, which included for all patients combination of glucocorticoid (budesonide or beclomethasone) and beta-adrenoceptor stimulant (formoterol). All 8 patients of the severe course group also received additional m-cholinergic receptor blocker (tiotropium bromide).

### 2.1. Flow Cytometry

Sample preparation and data calculation protocol were carried out according to the method developed and validated in the Laboratory of Molecular Immunology and described in detail earlier [21] based on a commercial BD QuantiBRITE PE kit (BD Biosciences).

Phenotyping was carried out in whole blood after lysing using BD FACS lysing solution (BD Biosciences, San Diego, CA, USA) following the manufacturer’s protocol. Samples were analyzed by flow cytometry on a FACSVerse flow cytometer (BD Biosciences) using the following monoclonal antibodies: anti-human CD3 V421, anti-human CD19 fluorescein isothiocyanate (FITC), anti-human CD8 allophycocyanin (APC)-cyanine (Cy)7, anti-human CD5-APC-Cy7, anti-human CD14 phycoerythrin (PE)-Cy7, anti-human CD25 FITC, anti-human CD127 (IL-7Rα) APC-Cy7, anti-human CD45RO FITC, anti-human CD45RA Pacific Blue, anti-human CD4 PE-Cy7 (all from BioLegend, San Diego, CA, USA); anti-human TNFR1-PE, anti-human TNFR2-PE, anti-human TNFR1-APC, and anti-human TNFR2-APC (R&D Systems, Minneapolis, MN, USA). Data analysis and calculation of fluorescence intensity parameters was performed using FacsDiva software (BD Biosciences).

Double labeling of samples with two antibody pairs (TNFR1-PE + TNFR2-APC and TNFR2-PE + TNFR1-APC) was performed to simultaneously quantify TNFR expression on different subsets (TNFR1+TNFR2-, TNFR1+TNFR2+, TNFR1-TNFR2+, and TNFR1-TNFR2- populations). Following flow cytometry, the number of TNFR1 receptors (for TNFR1+TNFR2- and TNFR1+TNFR2+ populations) in tubes containing TNFR2-PE and TNFR1-APC was calculated. The percentage of each population was determined as an average of two samples.

### 2.2. Statistical Analyses

Statistical analysis was performed using STATISTICA 7.0 software (StatSoft, Tulsa, OK, USA). Data were presented as medians and IQRs. Differences between independent samples (study subgroups) were assessed using the non-parametric Kruskal–Wallis test via multiple comparison of medians. Statistically significant predictors of the presence of BA and its severity were identified by univariate and multivariate logistic regression analysis. The indicators of co-expression and quantitative expression of TNF receptors, as well as the parameters of therapy received before hospitalization, and indicators of the severity of the disease were considered as predictors. For multivariate models, parameters were selected by stepwise reduction of statistically insignificant parameters. Quality parameters of the models were compared for all possible combinations of bivariate and trivariate models. For the final models characterized by the best quality parameters (with respect to R^2^ and the Akaike information criterion [22]), prognostic combinations based on patient baseline characteristics were visualized. Values of *p* < 0.05 were considered statistically significant.

## 3. Results

### 3.1. Co-Expression Patterns of TNFRs in Mononuclear Cell Subsets

The patients with BA demonstrated some significant differences in the expression of all the TNFRs in comparison with healthy donors (Figure 1).

In the total monocyte population, the expression profiles of TNFRs in patients with severe BA were similar to those of healthy volunteers, while more significant differences were observed among the patients with mild and moderate BA. An interesting difference was a distinct increase in the percentage of double-positive cells: 21.2% in healthy volunteers, 97% in the patients with mild BA (*p* = 0.152), 85.4% in the patients with moderate BA (*p* = 0.0037), and 33.8% in the patients with severe BA (*p* > 0.999). In addition, there was a reduction in the percentage of cells expressing only TNFR2: 57.3% in healthy volunteers, 2.2% in the patients with mild BA (*p* = 0.032), 4.4% in the patients with moderate BA (*p* = 0.0002), and 46.5% (*p* > 0.999) in the patients with severe BA.

In the patients with BA, the B cell population was characterized by changes in the percentage of cells expressing all the TNFRs. The most significant changes in the levels of individual expression and co-expression of TNFR1 were observed in the patients with mild BA: the percentage of cells exclusively expressing TNFR1 was increased almost tenfold, from 2.84% in healthy volunteers to 27.6% in the patients with BA (*p* = 0.0014), while the percentage of double-positive cells was slightly reduced. The most significant changes in the expression of TNFR2 were observed in the patients with moderate and severe BA, while the percentages of double-positive cells and cells exclusively expressing TNFR2 were reduced.

In the total T cell population, all the patients with BA had significantly higher percentages of cells exclusively expressing TNFR1: 3.46% in healthy volunteers, 24.87% in patients with mild BA (*p* = 0.027), 13.07% in patients with moderate BA (*p* = 0.005), and 22.1% in patients with severe BA (*p* = 0.034). All the patients with BA had lower percentages of double-negative (TNFR1-TNFR2-) cells compared with healthy volunteers; however, no statistically significant difference was found: 57.35% in healthy volunteers, 34.99% in patients with mild BA (*p* > 0.999), 44.98% in patients with moderate BA (*p* = 0.768), and 18.58% in patients with severe BA (*p* = 0.821). The most significant differences in TNFR1 expression were detected in the patients with severe BA, while the most significant differences in the co-expression of both receptors were detected in the patients with mild BA.

Among regulatory T cells, a higher percentage of double-negative cells was observed in the patients with severe BA compared with healthy volunteers (27% vs. 4.27%, *p* = 0.0024). This difference resulted primarily from the reduction in the percentage of cells exclusively expressing TNFR2.

The CD19+CD5+ B cell subset was characterized by a pronounced redistribution of TNFR expression among the patients with BA compared with healthy volunteers. Significant differences in the expression of TNFR2 were observed in the patients with severe BA compared with healthy volunteers: 75.45% in healthy volunteers, 30.62% in patients with mild BA (*p* = 0.038), 29.29% in patients with moderate BA (*p* = 0.051), and 54.5% in patients with severe BA (*p* = 0.042). The patients with mild and moderate BA also had significantly higher percentages of double-negative cells compared with healthy volunteers (*p* = 0.024 and *p* = 0.0019, respectively). There were almost no cells expressing TNFR1 in the patients with moderate and severe BA compared with healthy volunteers (*p* = 0.0057 and *p* = 0.0072, respectively).

### 3.2. Co-Expression of TNFR1 and TNFR2 on CD4+ Cells in Patients with Different Severities of BA and in Healthy Volunteers

The total pool of CD4+ positive cells was characterized by high percentages of cells exclusively expressing TNFR1 in all the patients with BA compared with healthy volunteers (Figure 1B). Conversely, a lower percentage of cells exclusively expressing TNFR2 was observed in the patients with moderate and severe BA compared with healthy volunteers.

The naïve T helper cell subset was characterized by lower percentages of double-positive cells in the patients with mild BA compared with healthy volunteers. In addition, there were higher percentages of cells exclusively expressing TNFR1 in the patients with severe BA compared with healthy volunteers.

The activated Th cell subset was characterized by lower percentages of cells exclusively expressing TNFR2 and high percentages of cells exclusively expressing TNFR1 in all the patients with BA compared with healthy volunteers. The patients with mild BA had higher percentages of double-positive cells compared with healthy volunteers.

The memory Th cell subset was characterized by higher percentages of cells exclusively expressing TNFR1 and lower percentages of cells expressing TNFR2 among all the patients with BA compared with healthy volunteers.

### 3.3. Co-Expression of TNFR1 and TNFR2 on CD8+ Cells in Patients with Different Severities of BA and in Healthy Volunteers

The total pool of CD8+ cells was characterized by lower percentages of cells exclusively expressing TNFR2 and higher percentages of cells expressing TNFR1 in all the patients with BA compared with healthy volunteers (Figure 1C). The patients with moderate and mild BA had higher percentages of double-positive cells. The patients with moderate BA had lower percentages of double-negative cells compared with healthy volunteers.

The naïve cytotoxic T cell subset was characterized by higher percentages of cells exclusively expressing TNFR1 among all the patients with BA compared with healthy volunteers. The patients with mild and severe BA had lower percentages of double-negative cells compared with healthy volunteers.

The activated cytotoxic T cell subset was characterized by higher percentages of cells exclusively expressing TNFR1 among all the patients with BA and lower percentages of cells expressing TNFR2 in the patients with mild and moderate BA compared with healthy volunteers.

The cytotoxic memory cell subset was characterized by a near-total absence of double-negative cells in the patients with mild and moderate BA compared with healthy volunteers. Healthy volunteers had almost no cells exclusively expressing TNFR1, while the percentage of these cells was higher in all the patients with BA. The patients with severe BA also had lower percentages of cells expressing TNFR2 compared with healthy volunteers.

### 3.4. Density of TNFR1 and TNFR2 Expression on Immune Cells in Patients with Different Severities of BA and in Healthy Volunteers

Significant differences were observed between the patients with BA and healthy volunteers in the average number of TNFRs per cell for the major immune cell subsets (Figure 2A). The number of TNFR1 molecules expressed by T cell, B cell, Tregs, CD5+ B cell, and CD14+ monocyte populations differed significantly in the BA patients compared with healthy volunteers. In all these cases, the levels of TNFR1 expression in healthy volunteers were higher than those of the patients with BA. 

Regarding the expression of the type 2 receptor, there was no similar clear trend for a one-sided change (decrease or increase) in the expression density in the BA patients compared with healthy volunteers. The T cell populations of the patients with moderate and severe BA had higher TNFR2 expression compared with those of healthy volunteers. The B cell population in the patients with severe BA was characterized by lower TNFR2 expression compared with that in healthy volunteers.

The total pool of CD4+ cells was characterized by lower numbers of TNFR1 and TNFR2 molecules in the patients with severe BA compared with healthy volunteers (Figure 2B). The number of TNFR2 molecules was higher in the patients with mild BA compared with healthy volunteers. The activated Th cell subset was characterized by low expression of TNFR1 in the patients with moderate and severe BA. The naïve cytotoxic Th subset was characterized by lower numbers of TNFR1 molecules in all the patients with BA, as well as lower numbers of TNFR2 molecules in the patients with moderate BA. The memory Th cell subset was characterized by low levels of TNFR1 and high levels of TNFR2 among all the patients with BA compared with healthy volunteers. The regulatory T cell subset was characterized by low levels of TNFR1 expression in all the patients with BA compared with healthy volunteers.

The total population of CD8+ cells was characterized by high TNFR1 expression in the patients with mild BA compared with healthy volunteers (Figure 2C). High expression levels of TNFR1 on naïve cytotoxic cells were observed in the patients with moderate and mild BA. In the patients with moderate BA, lower TNFR1 expression on the activated cytotoxic T cell subset and larger numbers of TNFR1 molecules on cytotoxic memory T cells were observed.

The total CD8+ cell pool expressed high TNFR2 levels in the patients with mild BA. In the activated cytotoxic cell subset, high TNFR2 levels were expressed in the patients with moderate BA. In the naïve cytotoxic cell and cytotoxic memory cell subsets, high TNFR2 expression levels were observed among all the patients with BA compared with healthy volunteers.

### 3.5. Regression Analysis to Identify Biomarkers of BA

The phenotyping data were included in univariate (Table 2) and multivariate (Table 3) logistic regression to assess their associations with the presence and severity of BA. The original univariate analysis included all 78 parameters of TNF receptor expression and co-expression in study populations, as well as data on treatment received and patient demographic characteristics. The identified statistically significant predictors are presented in the table. The final multivariate model included three parameters: the percentage of double-positive cells among CD19+CD5+ B cells, the average number of TNFR1 molecules on TNFR1-positive CD8+CD45RA+ cytotoxic naïve T cells, and the average number of TNFR1 molecules on TNFR1-positive CD4+ T helper cells. 

Figure 3 illustrates how the probability of developing BA increases (i) as the percentage of double-positive cells among CD19+CD5+ B cells decreases below 20% (and especially below 10%), (ii) as the number of TNFR1 molecules on CD8+CD45RA+ cells decreases below 1000 receptors per cell, and (iii) as the number of TNFR1 molecules on CD4+ cells increases above 600 to 800 receptors per cell.

## 4. Discussion

By comparing the patterns of co-expression of the TNFRs on the immune cells of patients with BA and healthy volunteers, we revealed distinct patterns of TNFR expression associated with BA. These TNFR expression differences were associated with different immune cell subsets and BA disease severity. To better understand the role of the ratio between TNFR1 and TNFR2 expression, we compared the numbers of receptors and the percentage of cells expressing receptors in major immune cell populations, including monocytes, T, and B cells, in healthy volunteers and patients with BA.

The assessment of TNFR co-expression on immune cell populations showed that the greatest differences were observed among monocytes and B cells. Significant differences in TNFR1 and TNFR2 expression were observed in monocytes. Interestingly, more than 90% of the cells in the patients with mild BA express both TNFR1 and TNFR2, while the percentage of double-positive cells was somewhat lower in the patients with moderate and severe BA. These observations were accompanied by smaller numbers of cell surface TNFR1 molecules in the patients with BA compared with healthy volunteers. This can be explained by two possible mechanisms. On the one hand, a decrease in membrane-bound forms of TNFR1 is usually associated with the shedding process and an increase in the level of soluble forms of this receptor, demonstrated in earlier studies [23], and reflecting a high level of systemic inflammation. On the other hand, this may be due to a change in the susceptibility of cells to the action of cytokines. Previous studies have shown that asthma severity is associated with changes in circulating monocyte subsets [24]. Moreover, TNFR1 expression levels are highest on classical monocytes, moderate on intermediate monocytes, and lowest on non-classical monocytes, whereas TNFR2 expression shows the opposite pattern. Thus, we can infer that different monocyte subsets integrate TNF signals in different manners to execute their functions [25]. Thus, the differences in co-expression patterns and numbers of TNFRs may be attributable to differences in monocyte subset composition in patients with BA of different severities.

In this study, the B cells of the patients with BA were characterized by higher percentages of double-negative cells, lower percentages of cells exclusively expressing TNFR2, and higher percentages of cells exclusively expressing TNFR1 compared with those of healthy volunteers. In addition, the B cells of the patients with BA had smaller numbers of both cell surface TNFR1 and TNFR2 molecules; the greatest differences were observed in the patients with moderate BA. Similar results were observed for the CD19+CD5+ B cell subpopulation. These data can be explained by the role of TNFR dysregulation in the pathogenesis of allergic diseases; in particular, reduced TNFR2 expression has been associated with allergy progression [19].

Although T-regulatory cells play an important role in the development and maintenance of inflammation in BA, we found a significant decrease in the expression density of type 1 receptors on them and a decrease in the proportion of cells carrying at least one of the types of receptors in severe disease. This is consistent with other studies that have shown that the TNF-mediated regulation of T-regulatory cells occurs through their induction by TNFR2-positive dendritic cells [9]. This can lead to a compensatory decrease in the level of TNFR1 expression on Tregs with increasing TNFR2 expression through a feedback loop leading to the formation of a potent suppressive phenotype in Treg [26], and it also emphasizes the importance of the cross-population interaction in the TNF receptor system regulation.

The assessment of the co-expression of TNFRs on T cell subsets showed that both the CD4+ and CD8+ cells of the patients with BA were characterized by the redistribution of TNFR expression and differences in the numbers of receptors on cells depending on the T cell subset and the severity of BA. In some cases, differences had opposite magnitudes for different subsets and for patients with different BA severity. Activated helper T cells have been implicated in the pathogenesis of BA and were previously shown to be characterized by the redistribution of TNFR co-expression [27]. In our study, this cell subset was characterized by lower percentages of cells exclusively expressing TNFR2 and higher percentages of cells exclusively expressing TNFR1 among all the patients with BA compared with healthy volunteers. The patients with mild BA showed more pronounced differences in TNFR expression among activated helper T cells and, additionally, had higher percentages of double-positive cells compared with healthy volunteers. TNFR2 expression is required for the activation of dormant T cells. The interaction between TNF-α and TNFR2 is required as co-stimulation during the T-cell-receptor-mediated activation of T cells and promotes the activation, differentiation, and survival of T cells [28]. Studies focused on BA showed that disruption of TNF-TNFR2 signaling aggravates airway inflammation, thus contributing to Th2 and Th17 cell polarization [16]. Hence, the low percentage of cells exclusively expressing TNFR2 and the higher percentage of cells expressing TNFR1 in the patients with BA compared with healthy volunteers observed in our study may be attributed to the presence of airway inflammation. Interestingly, the largest changes were observed in the patients with mild BA. Thus, changes in cellular immune function in patients with BA of different severities may be manifested through differences in TNFR co-expression. Additional research is needed as few studies have focused on TNFR expression in patients with BA of different severities.

Among all the patients with BA, all the subsets of CD8+ T cells were characterized by high percentages of cells exclusively expressing TNFR1 compared with healthy volunteers. The amount of TNFR2 on the subsets of CD8+ cells in the patients with BA is high or extremely high compared with that in healthy volunteers. Studies have shown that it is TNFR2 rather than TNFR1 that is the predominant TNF receptor upon the activation of CD8+ effector cells. Therefore, the direct effects of TNF-α on CD8+ T effector (Teff) cells are mainly mediated by TNFR2. It was reported that TNFR2 expression is involved in the activation of CD8+ Teff during certain phases of the immune response. Thus, TNFR2 both reduces the T cell activation threshold and ensures early costimulatory signals during T cell activation [29]. The activation of TNFR1 in TNFR2 knockout mice also contributes to the survival of CD8+ cells [30]. CD8+ cells play a key role in the development of severe steroid-resistant forms of BA [31]. The large number of TNFR2 molecules on the surface of CD8+ cells can be explained by the fact that co-stimulation is necessary for cell differentiation. The large percentage of cells exclusively expressing TNFR1 may also be related to the pathogenesis of BA.

Based on our analysis of the associations between BA severity and TNFR expression, we built predictive models. The predictors identified during mathematical modeling underline the significance of changes in receptor redistribution patterns during the development of the disease and reflect the clinical significance of these changes. The model aiming to discriminate patients with BA from healthy volunteers showed a high R^2^ value. The final multivariate model included three parameters: the percentage of double-positive cells among CD19+CD5+ B cells, the average number of TNFR1 molecules on TNFR1-positive CD8+CD45RA+ cytotoxic naïve T cells, and the average number of TNFR1 molecules on TNFR1-positive CD4+ T helper cells. TNFR1 expression was a key parameter associated with differences in immune cell frequencies that discriminate patients with BA from healthy individuals, as well as with disease severity. It is noteworthy that, of all the studied predictive models, none of the models containing parameters associated with changes in the type 2 receptor showed high rates of specificity or sensitivity. Despite the significant contribution of TNF and its receptors to the pathogenesis of bronchial asthma [13], our finding may indicate the predominant role of the TNFR1-mediated signaling pathways in the development and progression of the disease and explain why therapy of glucocorticoid-resistant bronchial asthma with soluble TNF blockers based on type 2 receptors had only partial efficiency [32,33]. Our data on the key role of type 1 receptors also support experimental research on the use of TNFR1 fragments to develop new therapeutic approaches to suppress asthma in experimental models [14].

## 5. Conclusions

Our data showed that TNFR co-expression and the average number of TNFR molecules on cells are crucial parameters for understanding cellular regulation both under normal and pathological conditions. Differences in TNFR co-expression and the number of TNFR molecules in different cell populations indicated that different immune cell types vary in their contributions to BA pathogenesis. Moreover, differences in these parameters associated with disease severity may reflect changes in the composition of immune cell populations and functional cellular responses. Overall, these findings are helpful to characterize the unique TNFR expression profile in the immune cells of patients with BA and may be helpful in applied diagnostic testing of BA.

## Figures and Tables

**Figure 1 cells-11-01736-f001:**
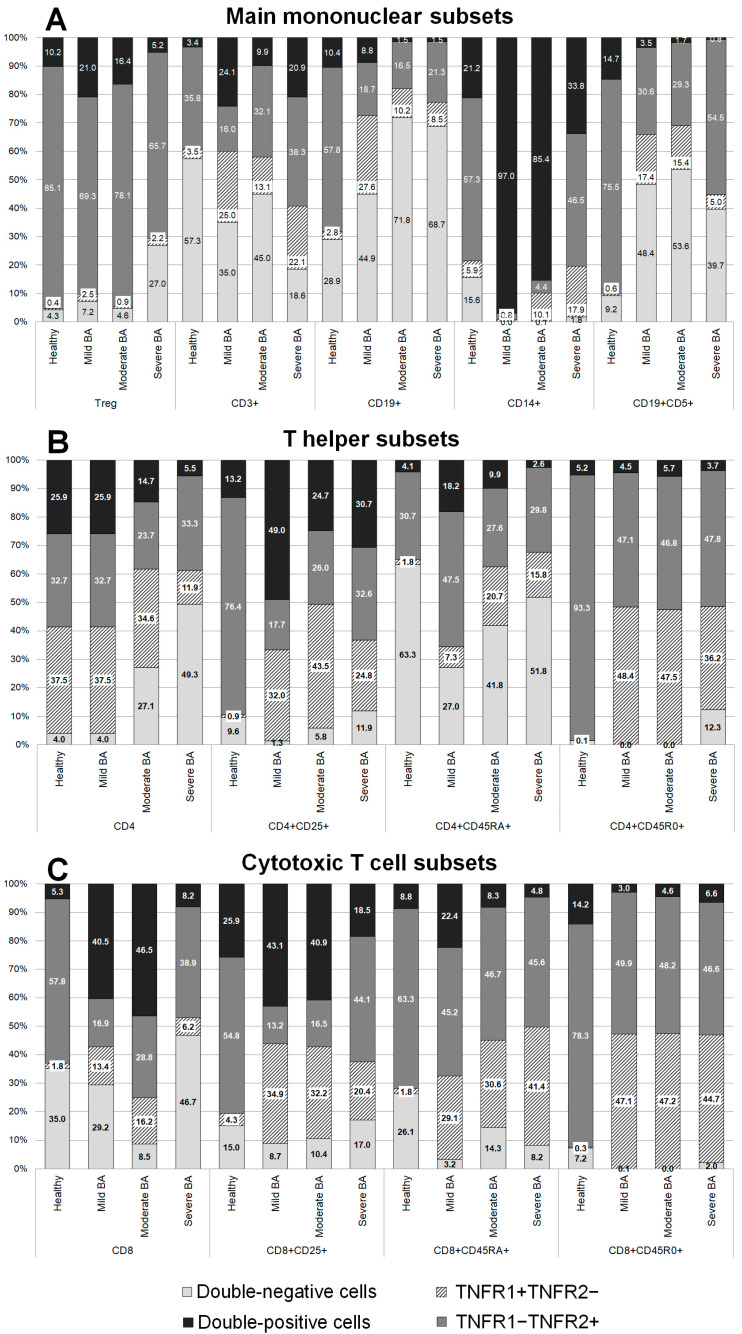
Distribution of TNFR1 and TNFR2 co-expression among main mononuclear subsets (**A**), among T helper subsets (**B**), and among cytotoxic T cell subsets (**C**).

**Figure 2 cells-11-01736-f002:**
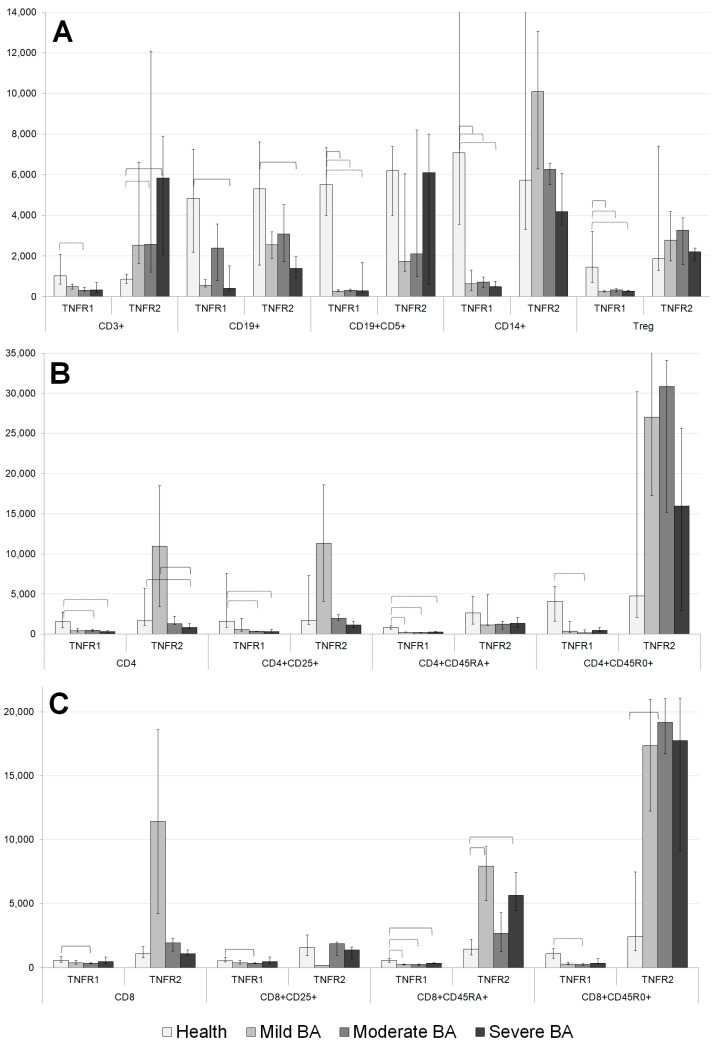
Expression levels of TNFRs in main mononuclear cell subsets (**A**), on CD4+ cells (**B**), and on CD8+ cells (**C**). Solid bracket indicates significant difference with *p* < 0.05; dotted bracket indicates significant difference with *p* < 0.005 according to Kruskal–Wallis median test.

**Figure 3 cells-11-01736-f003:**
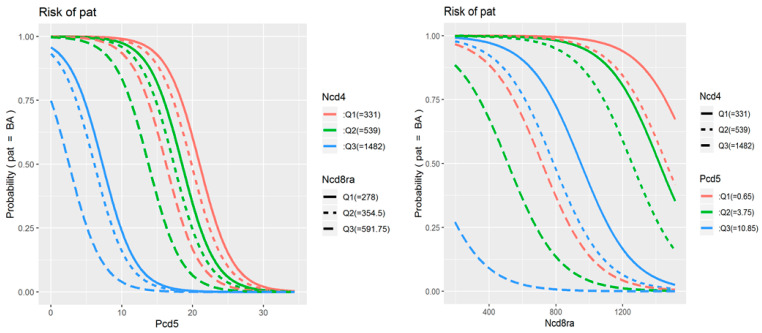
Probability of BA accessed in multivariate logistic regression analysis depending on achieved threshold levels on each of 3 parameters. Ncd4 indicates average number of TNFR1 molecules on TNFR1-positive CD4+ T helper cells; Ncd8ra indicates average number of TNFR1 molecules on TNFR1-positive CD8+CD45RA+ naïve cytotoxic T cells; Pcd5 indicates percentage of double-positive cells among CD19+CD5+ B cells. Q1, Q2, Q3 indicate three threshold levels on each of the parameters.

**Table 1 cells-11-01736-t001:** Baseline characteristics of study participants.

Indicator	Healthy Volunteers (n = 46)	Patients with BA (n = 22)
Female gender, n (%)	30 (65.2%)	16 (72.7%)
Age, median (IQR)	36.5 (30–54)	46 (32–49)
**Baseline therapy**		
ICS/LABA	NA	13 (59%)
ICS/LABA and tiotropium bromide	NA	3 (14%)
ICS/LABA, tiotropium bromide, and LTRA	NA	6 (27%)
**Inpatient therapy**		
Parenteral CS	NA	6 (27%)
Oral and parenteral CS	NA	2 (9%)
CS and LTRA	NA	12 (55%)
Oral, parenteral CS, and LTRA	NA	2 (9%)
**BA severity, n (%)**		
Mild (FEV1 > 80%)	NA	4 (18.1%)
Moderate (FEV1 50–79%)	NA	10 (45.5%)
Severe (FEV1 49–30%)	NA	8 (36.4%)

Abbreviations: BA, bronchial asthma; IQR, interquartile range; ICS, inhaled corticosteroids; LABA, long-acting beta agonist; LTRA, leukotriene receptor antagonist; CS, corticosteroid; FEV1, forced expiratory volume in 1 s.

**Table 2 cells-11-01736-t002:** Results of univariate logistic regression analysis.

Parameter	OR (2.5–97.5%)	*p*-Value
Percentage of TNFR1+ cells among CD4+CD25+ cells	1.484 (1.268–1.883)	<0.001
Percentage of TNFR1+ cells among CD4 cells	1.391 (1.212–1.738)	<0.001
Percentage of TNFR1+ cells among CD4+CD45R0+ cells	1.333 (1.169–1.728)	0.002
Percentage of TNFR1+ cells among CD8+CD45RA+ cells	1.298 (1.184–1.497)	<0.001
Percentage of TNFR1+ cells among CD19+CD5+ B cells	1.288 (1.153–1.514)	<0.001
Percentage of TNFR1+ cells among CD8+ cells	1.23 (1.121–1.39)	<0.001
Percentage of TNFR1+ cells among CD8+CD45R0+ cells	1.224 (1.132–1.415)	<0.001
Percentage of double-positive cells among CD19+CD5+ B cells	0.819 (0.739–0.888)	<0.001
Percentage of TNFR1+ cells among CD8+CD25+ cells	1.194 (1.121–1.298)	<0.001
Age (years)	1.034 (1.004–1.066)	0.029
Mean number of TNFR1 on CD4+CD45RA+ double-positive (TNFR1+TNFR2+) cells	0.992 (0.988–0.995)	<0.001
Mean number of TNFR1 on CD8+CD45RA+ cells with TNFR1	0.993 (0.99–0.996)	<0.001
Mean number of TNFR1 on CD4+CD45RA+ cells with TNFR1	0.995 (0.993–0.997)	<0.001
Mean number of TNFR1 on CD4+ cells with TNFR1	0.996 (0.994–0.998)	<0.001
Mean number of TNFR1 on CD4+CD25+ double-positive (TNFR1+TNFR2+) cells	0.997 (0.995–0.998)	<0.001

Abbreviations: TNFR, tumor necrosis factor receptor.

**Table 3 cells-11-01736-t003:** Results of multivariate logistic regression analysis.

Patients with BA vs. Health Volunteers	OR	2.5%	97.5%	*p*-Value
Percentage of double-positive cells among CD19+CD5+ B cells	1.533723	1.2	2.62	0.013
Average number of TNFR1 molecules on TNFR1-positive CD8+CD45RA+ naïve cytotoxic T cells	1.006475	1	1.02	0.042
Average number of TNFR1 molecules on TNFR1-positive CD4+ T helper cells	1.005105	1	1.01	0.018

Abbreviations: BA, bronchial asthma; TNFR, tumor necrosis factor receptor.

## Data Availability

The data presented in this study are available on request from the corresponding author.

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
