# Peer review of "Redistribution of TNF Receptor 1 and 2 Expression on Immune Cells in Patients with Bronchial Asthma"

_cells, 2022, doi:10.3390/cells11111736_

Round 1
Reviewer 1 Report
The article is deals with the redistribution of TNF receptor 1 and 2 expression on immune cells in patients with bronchial asthma.
I have some comments:
line 18: “control” should be replaced on “comparison”. Control group is used during the experiments on animals;
line 45: level should be replaced on levels;
line 66: “nominally” should be deleted (from healthy volunteers);
Methods should be described more carefully.2.1. should be titled “Patients”. It is necessary to describe the Baseline therapy and Inpatient therapy by subgroups (mild, moderate, severe). Sample preparation for cytometry should be described in more detail.
Figure1 “health” should be replaced on “healthy”.
It is desirable to analyze does the kind of therapy influence on the redistribution of TNF receptors.
It is not clear how predictive models can be applied in diagnostic testing of BA, needs to be clarified.
The article is deals with the redistribution of TNF receptor 1 and 2 expression on immune cells in patients with bronchial asthma.
I have some comments:
line 18: “control” should be replaced on “comparison”. Control group is used during the experiments on animals;
line 45: level should be replaced on levels;
line 66: “nominally” should be deleted (from healthy volunteers);
Methods should be described more carefully.2.1. should be titled “Patients”. It is necessary to describe the Baseline therapy and Inpatient therapy by subgroups (mild, moderate, severe). Sample preparation for cytometry should be described in more detail.
Figure1 “health” should be replaced on “healthy”.
It is desirable to analyze does the kind of therapy influence on the redistribution of TNF receptors.
It is not clear how predictive models can be applied in diagnostic testing of BA, needs to be clarified.
Author Response
line 18: “control” should be replaced on “comparison”. Control group is used during the experiments on animals;
The term has been changed. We have also replaced throughout the text all mentions of “healthy controls” to “healthy volunteers”.
line 45: level should be replaced on levels;
The typo has been corrected.
line 66: “nominally” should be deleted (from healthy volunteers);
Paraphrasing has been done in two places.
Methods should be described more carefully.2.1. should be titled “Patients”. It is necessary to describe the Baseline therapy and Inpatient therapy by subgroups (mild, moderate, severe). Sample preparation for cytometry should be described in more detail.
We have added a detailed description of the treatment received by patients of different groups before admission to the current hospitalization. However, since blood sampling for analysis was carried out before the correction and start of the modified therapy in the hospital, this information could not affect the level of receptor expression and was not taken into account in the current article. Clarifications have been added to the text.
Sample preparation and data calculation protocol were carried out according to the method developed and validated in our laboratory and described in detail earlier. This information has been added to the Methods section.
Figure1 “health” should be replaced on “healthy”.
Captions have been changed on all figures
It is desirable to analyze does the kind of therapy influence on the redistribution of TNF receptors.
Since basic therapy differed only in the group with a severe course of the disease, and correlation analysis and regression analysis did not show significant associations for this group, this aspect was not covered in detail in the article. Added explanatory information to the methods and results sections.
It is not clear how predictive models can be applied in diagnostic testing of BA, needs to be clarified.
The value of our mathematical model lies primarily not in its use as a diagnostic algorithm, but in confirmation of the significance of changes in receptor redistribution patterns during the development of the disease and reflects the clinical significance of these changes. We have added an appropriate explanation to the discussion.
Reviewer 2 Report
1.It is recommended to list whether the patient with bronchial asthma have other comorbidities that will affect the laboratory blood values, especially immune cells.
- Although there are differed significantly in the results, but the number of the experimental group is small (22 patients with bronchial asthma), and divided into three groups (severe (n=8), moderate (n=10), and mild (n=4)). It is recommended to increase the sample size of experimental groups.
- Conclusions demonstrated that expression differences were associated with disease severity. The disease severity of the patients was differentiated only by FEV (shown in Table 1). It is recommended to use statistical methods to understand the association between drugs, disease severity and various blood values in order to clarify whether disease severity and laboratory blood values, especially immune cells, are affected by drugs.
- Among all references (32), there are 43.75% (14) over 5 years. It is recommended that references are within 5 years
Author Response
1.It is recommended to list whether the patient with bronchial asthma have other comorbidities that will affect the laboratory blood values, especially immune cells.
The criterion for inclusion of patients in the study was the absence of other immune-mediated conditions and diseases that could affect the redistribution of receptors on the surface of immunocompetent cells. We have included this information in the Methods section.
- Although there are differed significantly in the results, but the number of the experimental group is small (22 patients with bronchial asthma), and divided into three groups (severe (n=8), moderate (n=10), and mild (n=4)). It is recommended to increase the sample size of experimental groups.
Due to rather strict restrictions on the inclusion of patients to exclude the influence of comorbidities on the co-expression profile, we included in this study all patients treated at the center for 3 years of the project, and expanding the sample is currently not possible due to the end of research. However, the established statistically significant differences confirm the sufficiency of the sample for our conclusions.
- Conclusions demonstrated that expression differences were associated with disease severity. The disease severity of the patients was differentiated only by FEV (shown in Table 1). It is recommended to use statistical methods to understand the association between drugs, disease severity and various blood values in order to clarify whether disease severity and laboratory blood values, especially immune cells, are affected by drugs.
All proposed statistical analyzes were carried out and demonstrated the greatest predictive significance of the receptor expression indicators, which was associated with the homogeneity of the patient sample (and was included in the study plan to exclude the influence of other parameters). Added clarifying phrases to the Methods and Results section.
- Among all references (32), there are 43.75% (14) over 5 years. It is recommended that references are within 5 years
Despite the important role of the cytokine itself in the development and course of bronchial asthma (and, consequently, the inherent importance of the role of its receptors in this pathology), the study of the expression of TNF receptors in the disease has been poorly studied. A systematic literature search showed that over the past 7 years (since 2015) only 3 original articles have been published on the study of this issue by other scientific teams. We have added references to all of these articles, and also replaced references to reviews on general issues of studying receptors and the role of TNF in bronchial asthma with the most recent ones. We had to let few articles older than 2017 stay as fundamental research in this area, which have no later analogues, but their number was minimized to 8/33 (24%).
Round 2
Reviewer 1 Report
Thank you
Author Response
Thank you again for evaluating our work.
Reviewer 2 Report
- Please confirm the indications on lines 38 to 40. [2, 3 Gough P, Myles IA. Tumor Necrosis Factor Receptors: Pleiotropic Signaling Complexes and Their Differential Effects. Front Immunol. 2020 Nov 25;11:585880. doi: 10.3389/fimmu.2020.585880. PMID: 33324405; PMCID: PMC7723893.].
- In the abstract and the title item of Table 1, the total number of healthy volunteers is 43, but the total number of healthy volunteers in the method and the content of Table 1 is 46. Please confirm which is correct, and please check all the data in the full text.
For example:
Abstract: data from 43 healthy volunteers (HV) were accessed
Table 1: Healthy volunteers (n=43)
Method: A total of 46 healthy volunteers aged 18 to 77 years (median, 36.5 years; interquartile range [IQR], 30 to 54 years) were enrolled in the study: 16 men (34.8%) and 30 women (65.2%).
Table 1: Female gender, n (%) 30 (65.2%)ï¼›Age, median (IQR) 36.5 (30–54)
- Please unify the description of the numbers in the full text, such as eight men (27.3%) and 16 (72.7%) women.
- Please check: Venous blood samples (6 mL) were collected from the ulnar vein under fasting conditions into vacuum tubes containing ethylenediaminetetraacetic acid (Greiner Bio-One GmbH, Kremsmünster, Austria) using aseptic technique before the correction of basis therapy.
- In this paper, FEV1 is used as a classification of disease severity, and it is only presented in Table 1. It is recommended to briefly describe the collection process of FEV1, such as collection time, instruments and methods, etc.
- Are healthy volunteers tested for FEV1? Are all FEV1 within the normal range?
Author Response
1. Please confirm the indications on lines 38 to 40. [2, 3 Gough P, Myles IA. Tumor Necrosis Factor Receptors: Pleiotropic Signaling Complexes and Their Differential Effects. Front Immunol. 2020 Nov 25;11:585880. doi: 10.3389/fimmu.2020.585880. PMID: 33324405; PMCID: PMC7723893.].
We have removed the text of reference 3 and left the link only in the main text.
2. In the abstract and the title item of Table 1, the total number of healthy volunteers is 43, but the total number of healthy volunteers in the method and the content of Table 1 is 46. Please confirm which is correct, and please check all the data in the full text.
For example:
Abstract: data from 43 healthy volunteers (HV) were accessed
Table 1: Healthy volunteers (n=43)
Method: A total of 46 healthy volunteers aged 18 to 77 years (median, 36.5 years; interquartile range [IQR], 30 to 54 years) were enrolled in the study: 16 men (34.8%) and 30 women (65.2%).
Table 1: Female gender, n (%) 30 (65.2%)ï¼›Age, median (IQR) 36.5 (30–54)
Thank you for noticing the typo in the abstract and table 1, we have corrected the mention of 46 patients during the text.
3. Please unify the description of the numbers in the full text, such as eight men (27.3%) and 16 (72.7%) women.
We have unified the order of mentioning percentage during the text.
4. Please check: Venous blood samples (6 mL) were collected from the ulnar vein under fasting conditions into vacuum tubes containing ethylenediaminetetraacetic acid (Greiner Bio-One GmbH, Kremsmünster, Austria) using aseptic technique before the correction of basis therapy.
We have splited the sentence in two for simplicity and corrected the mistake.
5. In this paper, FEV1 is used as a classification of disease severity, and it is only presented in Table 1. It is recommended to briefly describe the collection process of FEV1, such as collection time, instruments and methods, etc.
Thank you for this clarification, we have added a detailed description of the procedure in the methods section.
6. Are healthy volunteers tested for FEV1? Are all FEV1 within the normal range?
Selection of healthy volunteers was carried out after basic check-up including an assessment of cardiac and respiratory functions. FEV1 measurement is not included in the standard donor health assessment protocol in Russia, however, the examination by the therapist was successfully completed. We have added a clarifying phrase to the Methods section.